# DISENTANGLING THE LINK BETWEEN IMAGE STATISTICS AND HUMAN PERCEPTION

## ABSTRACT

In the 1950s, Barlow and Attneave hypothesised a link between biological vision and information maximisation. Following Shannon, information was defined using the probability of natural images. A number of physiological and psychophysical phenomena have been derived ever since from principles like info-max, efficient coding, or optimal denoising. However, it remains unclear how this link is expressed in mathematical terms from image probability. First, classical derivations were subjected to strong assumptions on the probability models and on the behaviour of the sensors. Moreover, the direct evaluation of the hypothesis was limited by the inability of the classical image models to deliver accurate estimates of the probability. In this work we directly evaluate image probabilities using an advanced generative model for natural images, and we analyse how probability-related factors can be combined to predict the sensitivity of state-of-the-art subjective image quality metrics, a proxy for human perception. We use information theory and regression analysis to find a simple model that when combining just two probability-related factors achieves 0.77 correlation with subjective metrics. This probability-based model is psychophysically validated by reproducing the basic trends of the Contrast Sensitivity Function, its suprathreshold variation, and trends of the Weber-law and masking.

## 1 INTRODUCTION

One long standing discussion in artificial and human vision is about the principles that should drive sensory systems. One example is Marr and Poggio's functional descriptions at the (more abstract) *Computational Level* of vision (Marr & Poggio, 1976). Another is Barlow's *Efficient Coding Hypothesis* (Attneave, 1954; Barlow, 1961), which suggests that vision is just an optimal information-processing task. In modern machine learning terms, the classical optimal coding idea qualitatively links the probability density function (PDF) of the images with the behaviour of the sensors.

An indirect research direction explored this link by proposing design principles for the system (such as infomax) to find optimal transforms and compare the features learned by these transforms and the ones found in the visual system, e.g. receptive fields or nonlinearities (Olshausen & Field, 1996; Bell & Sejnowski, 1997; Simoncelli & Olshausen, 2001; Schwartz & Simoncelli, 2001; Hyvärinen et al., 2009). This indirect approach, which was popular in the past due to limited computational resources, relies on gross approximations of the image PDF and on strong assumptions about the behaviour of the system. The set of PDF-related factors (or surrogates of the PDF) that were proposed as explanations of the behaviour in the above indirect approach is reviewed below in Sec. 2.2.

In contrast, here we propose a direct relation between the behaviour (sensitivity) of the system and the PDF of natural images. Following the preliminary suggestions in (Hepburn et al., 2022) about this direct relation, we rely on two elements. On the one hand, we use recent *generative models* that represent large image datasets better than previous PDF approximations and provide us with an accurate estimate of the probability at query points (van den Oord et al., 2016; Salimans et al., 2017). Whilst these statistical models are not analytical, they allow for sampling and log-likelihood prediction. They also allow us to compute gradients of the probability, which, as reviewed below, have been suggested to be related to sensible vision goals.

On the other hand, recent measures of *perceptual distance* between images have recreated human opinion of subjective distortion to a great accuracy (Laparra et al., 2016; Zhang et al., 2018; Hep-

burn et al., 2020; Ding et al., 2020). Whilst being just approximations to human sensitivity, the sensitivity of the perceptual distances is a convenient computational description of the main trends of human vision that should be explained from scene statistics. These convenient proxies are important because (1) testing relations between the actual sensitivity and probability in *many* points and directions of the image space is not experimentally feasible, and (2) once an approximate relation has been established the metric can be used to identify discriminative points and directions where one should measure actual (human) sensitivities.

In this work, we identify the more relevant probabilistic factors that may be behind the non-Euclidean behaviour of perceptual distances and propose a simple expression to predict the perceptual sensitivity from these factors. First, we empirically show the relationship between the sensitivity of the metrics and different functions of the probability using conditional histograms. Then, we analyse this relationship quantitatively using mutual information, factor-wise and considering groups of factors. After, we use different regression models to identify a hierarchy in the factors that allow us to propose analytic relationships for predicting perceptual sensitivity. Finally, we perform an ablation analysis over the most simple closed-form expressions and select some solutions to predict the sensitivity given the selected functions of probability.

## 2 BACKGROUND AND PROPOSED METHODS

In this section, we first recall the description of visual behaviour: the *sensitivity of the perceptual distance* (Hepburn et al., 2022), which is the feature to be explained by functions of the probability. Then we review the probabilistic factors that were proposed in the past to be related to visual perception, but had to be considered indirectly. Finally, we introduce the tools to compute both behaviour and statistical factors: (i) the perceptual distances, (ii) the probability models, and (iii) how variations in the image space (distorted images) are chosen.

### 2.1 THE PROBLEM: PERCEPTUAL SENSITIVITY

Given an original image $\mathbf{x}$ and a distorted version $\tilde{\mathbf{x}}$, full-reference *perceptual distances* are models, $D_p(\mathbf{x}, \tilde{\mathbf{x}})$, that accurately mimic the human opinion about the subjective difference between them. In general, this perceived distance, highly depends on the particular image analysed and on the direction of the distortion. These dependencies make $D_p$ distinctly different from Euclidean metrics like the Root Mean Square Error (RMSE), $||\mathbf{x} - \tilde{\mathbf{x}}||_2$, which does not correlate well with human perception (Wang & Bovik, 2009). This characteristic dependence of $D_p$ is captured by the *directional sensitivity of the perceptual distance* (Hepburn et al., 2022):

$$S(\mathbf{x}, \tilde{\mathbf{x}}) = \frac{D_p(\mathbf{x}, \tilde{\mathbf{x}})}{||\mathbf{x} - \tilde{\mathbf{x}}||_2} \tag{1}$$

This is actually the numerical version of the directional derivative for a given $D_p$, and we will refer to it as the *perceptual sensitivity* throughout the work. In this approximation of human sensitivity using metrics, the increase of $D_p$ as $\tilde{\mathbf{x}}$ departs from $\mathbf{x}$ is conceptually equivalent to the empirical transduction functions that can be psychophysically measured in humans (e.g. by using maximum likelihood difference scaling Knoblauch & Maloney (2008)) along the direction $\mathbf{x} - \tilde{\mathbf{x}}$. While specific directions have special meaning, e.g. distortions that alter contrast or brightness, here we are interested in the general behaviour of the sensitivity of perceptual metrics for different images, $\mathbf{x}$.

### 2.2 PREVIOUS PROBABILISTIC EXPLANATIONS OF VISION

There is a rich literature trying to link the properties of visual systems with the PDF of natural images. Principles and PDF-related factors that have been proposed in the past include (Table 1):

**(i) Infomax for *regular images*.** Transmission is optimised by reducing the redundancy, as in principal component analysis (Hancock et al., 1992; Buchsbaum & Gottschalk, 1983), independent components analysis (Olshausen & Field, 1996; Bell & Sejnowski, 1997; Hyvärinen et al., 2009), and, more generally, in PDF equalisation (Laughlin, 1981) or factorisation (Malo & Laparra, 2010). Perceptual discrimination has been related to transduction functions obtained from the cumulative density Wei & Stocker (2017), as in PDF equalisation.

Table 1: Probabilistic factors that have been proposed to predict sensitivity. A general problem of the classical literature (in Sec. 2.2) is that direct estimation of the probability at arbitrary images $p(\mathbf{x})$ was not possible. $\mu(\mathbf{x})$ specifies the mean of $\mathbf{x}$, $\Sigma(\mathbf{x})$ is the covariance of $\mathbf{x}$ and $B$ is a mixing matrix.

| (i) | (ii) | | (iii) | (iv) | (v) | (vi) |
|---|---|---|---|---|---|---|
| Information Transmission | Internal Noise Limited Resolution | Acquisition Noise Denoising | Surprise | Signal Average First Eigenvalue | Signal Spectrum All Eigenvalues | Marginal Laplacian Marginal nonlinearity |
| $p(\mathbf{x})$ | $p(\mathbf{x})^{\frac{1}{3}}$ | $\frac{\nabla_{\mathbf{x}} p(\mathbf{x})}{p(\mathbf{x})}$ | $p(\mathbf{x})^{-1}$ | $\mu(\mathbf{x})$ | $\frac{1}{\mu(\mathbf{x})}\Sigma(\mathbf{x})$ | $\frac{1}{\mu(\mathbf{x})} B \cdot \log(\lambda) \cdot B^\top$ |
| $\log(p(\mathbf{x}))$ | $\frac{1}{3}\log(p(\mathbf{x}))$ | $J(\mathbf{x}) = \nabla_{\mathbf{x}}\log(p(\mathbf{x}))$ | $-\log(p(\mathbf{x}))$ | $\log(\mu(\mathbf{x}))$ | $\frac{1}{\mu(\mathbf{x})} B \cdot \lambda \cdot B^\top$ | $\int_{\mathbf{x}}^{\hat{\mathbf{x}}} \log(p(\mathbf{x}')) \, d\mathbf{x}'$ |

**(ii) Optimal representations for *regular images* in noise**. Noise may occur either at the front-end sensors (Miyasawa, 1961; Atick, 1992; Atick et al., 1992) (optimal denoising), or at the internal response (Lloyd, 1982; Twer & MacLeod, 2001) (optimal quantisation, or optimal nonlinearities to cope with noise). While in denoising the solutions depend on the derivative of the log-likelihood (Raphan & Simoncelli, 2011; Vincent, 2011), optimal quantisation is based on resource allocation according to the PDF after saturating non linearities (Twer & MacLeod, 2001; Macleod & von der Twer, 2003; Seriès et al., 2009; Ganguli & Simoncelli, 2014). Bit allocation and quantisation according to nonlinear transforms of the PDF has been used in perceptually optimised coding (Macq, 1992; Malo et al., 2000). In fact, both factors considered above (infomax and quantisation) have been unified in a single framework where the representation is driven by the PDF raised to certain exponent (Malo & Gutiérrez, 2006; Laparra et al., 2012; Laparra & Malo, 2015).

**(iii) Focus on *surprising images* (as opposed to regular images)**. This critically different factor (surprise as opposed to regularities) has been suggested as a factor driving sensitivity to color (Gegenfurtner & Rieger, 2000; Wichmann et al., 2002), and in visual saliency (Bruce & Tsotsos, 2005). In this case, the surprise is described by the inverse of the probability (as opposed to the probability) as in the core definition of information (Shannon, 1948).

**(iv) Energy (first moment, mean or average, of the signal)**. Energy is the obvious factor involved in sensing. In statistical terms, the first eigenvalue of the manifold of a class of images represents the average luminance of the scene. The consideration of the nonlinear brightness-from-luminance response is a fundamental law in visual psychophysics (the Weber-Fechner law (Weber, 1846; Fechner, 1860)). It has statistical explanations related to the cumulative density (Laughlin, 1981; Malo & Gutiérrez, 2006; Laparra et al., 2012) and using empirical estimation of reflectance (Purves et al., 2011). Adaptivity of brightness curves (Whittle, 1992) can only be described using sophisticated non-linear architectures (Hillis & Brainard, 2005; Martinez et al., 2018; Bertalmío et al., 2020).

**(v) Structure (second moment, covariance or spectrum, of the signal**. Beyond the obvious *energy*, vision is about understanding the *spatial structure*. The simplest statistical description of the structure is the covariance of the signal. The (roughly) stationary invariance of natural images implies that the covariance can be diagonalized in Fourier like-basis, $\Sigma(\mathbf{x}) = B \cdot \lambda \cdot B^\top$ (Clarke, 1981), and that the spectrum of eigenvalues in $\lambda$ represents the average Fourier spectrum of images. The magnitude of the sinusoidal components compared to the mean luminance is the concept of *contrast* (Michelson, 1927; Peli, 1990), which is central in human spatial vision. Contrast thresholds have a distinct bandwidth (Campbell & Robson, 1968), which has been related to the spectrum of natural images (Atick et al., 1992; Gomez-Villa et al., 2020; Li et al., 2022).

**(vi) Heavy-tailed marginal PDFs in transformed domains** were used by classical generative models of natural images in the 90's and 00's (Simoncelli, 1997; Malo et al., 2000; Hyvärinen et al., 2009; van den Oord & Schrauwen, 2014) and then these marginal models were combined through mixing matrices (either PCA, DCT, ICA or wavelets), conceptually represented by the matrix $B$ in the last column of Table 1. In this context, the response of a visual system (according to PDF equalisation) should be related to non-linear saturation of the signal in the transform domain using the cumulative functions of the marginals. These explanations (Schwartz & Simoncelli, 2001; Malo & Gutiérrez, 2006) have been given for the adaptive nonlinearities that happen in the wavelet-like representation in the visual cortex (Heeger, 1992; Carandini & Heeger, 2012), and also to define perceptual metrics (Daly, 1990; Teo & Heeger, 1994; Malo et al., 1997; Laparra et al., 2010; 2016).

## 2.3 OUR PROPOSAL

Here we revisit the classical principles in Table 1 to propose a *direct* prediction of the sensitivity of state-of-the-art perceptual distances from the image PDF. The originality consists on the *direct* com-

putation of $p(\mathbf{x})$ through current generative models as opposed to the indirect approaches taken in the past to check the Efficient Coding Hypothesis (when direct estimation of $p(\mathbf{x})$ was not possible). In our prediction of the perceptual sensitivity we consider the following PDF-related factors:

$$\log(p(\mathbf{x})) \, , \, \log(p(\tilde{\mathbf{x}})) \, , \, ||J(\mathbf{x})|| \, , \, ||J(\tilde{\mathbf{x}})|| \, , (\mathbf{x} - \tilde{\mathbf{x}})^{\top} \cdot J(\mathbf{x}) \, , \, \mu(\mathbf{x}) \, , \quad \sigma(\mathbf{x}) \, , \, \int_{\mathbf{x}}^{\tilde{\mathbf{x}}} \log(p(\mathbf{x}')) \, d\mathbf{x}' \quad (2)$$

where $\mathbf{x}$ is the image, $\tilde{\mathbf{x}}$ is the distorted image, $J(\mathbf{x})$ is the (vector) gradient of the log-likelihood at $\mathbf{x}$. The standard deviation of the image, $\sigma(\mathbf{x})$ (as a single element of the covariance $\Sigma(\mathbf{x})$), together with $\mu(\mathbf{x})$ capture the concept of RMSE contrast (Peli, 1990) (preferred over Michelson contrast (Michelson, 1927) for natural stimuli), and the integral takes the log-likelihood that can be computed from the generative models and accumulates it along the direction of distortion, qualitatively following the idea of cumulative responses proposed in equalisation methods (Laughlin, 1981; Malo & Gutiérrez, 2006; Laparra et al., 2012; Laparra & Malo, 2015)(Wei & Stocker, 2017).

## 2.4 Representative perceptual distances

The most successful perceptual distances can be classified in four big families: **(i) Physiological-psychophysical architectures.** These include (Daly, 1990; Watson, 1993; Teo & Heeger, 1994; Malo et al., 1997; Laparra et al., 2010; Martinez et al., 2019; Hepburn et al., 2020) and in particular it includes NLPD (Laparra et al., 2016), which consists of a sensible filterbank of biologically meaningful receptive fields and the canonical Divisive Normalization used in neuroscience (Carandini & Heeger, 2012). **(ii) Descriptions of the image structure.** These include the popular SSIM (Wang et al., 2004), its (improved) multiscale version MS-SSIM (Wang et al., 2003), and recent version using deep learning: DISTS (Ding et al., 2020). **(iii) Information-theoretic measures.** These include measures of transmitted information such as VIF (Sheikh & Bovik, 2006; Malo et al., 2021), and recent measures based on enforcing informational continuity in frames of natural sequences, such as PIM (Bhardwaj et al., 2020). **(iv) Regression models:** generic deep architectures used for vision tasks retrained to reproduce human opinion on distortion as LPIPS (Zhang et al., 2018).

Here we use representative examples of the four families: MS-SSIM, NLPD, DISTS, PIM, and LPIPS. Table 3 in Appendix A illustrates the performance of these measures in reproducing human opinion. Fig. 1 visually demonstrates the image-dependent sensitivity to noise, effectively captured by the representative $D_p$ measure NLPD, in contrast to the less effective Euclidean distance, RMSE.

## 2.5 A convenient generative model

Our proposal requires a probability model that is able to give an accurate estimate of $p(\mathbf{x})$ at arbitrary points (images) $\mathbf{x}$, so that we can study the sensitivity-probability relation at *many* images. In this work we rely on PixelCNN++ (Salimans et al., 2017), since it is trained on CIFAR10 (Krizhevsky et al., 2009), a dataset made up of small colour images of a *wide range* of natural scenes. Pixel-CNN++ is an autoregressive model that fully factorises the PDF over all pixels in an image, where the conditional distributions are parameterised by a convolutional neural network. PixelCNN++ also achieves the lowest negative log-likelihood of the test set, meaning the PDF is accurate in this dataset. This choice is convenient for consistency with previous works (Hepburn et al., 2022),

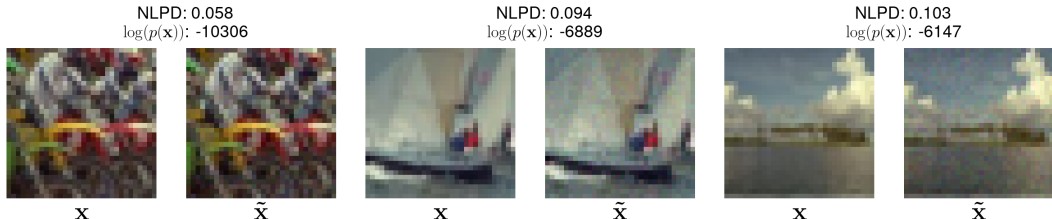

Figure 1: **The concept: visual sensitivity may be easy to predict from image probability.** Different images are corrupted by uniform noise of the same energy (on the surface of a sphere around $\mathbf{x}$ of Euclidean radius $\epsilon = 1$), with the same RMSE = 0.018 (in [0, 1] range). Due to *visual masking* (Legge & Foley, 1980), noise is more visible, i.e. human sensitivity is bigger, for smooth images. This is consistent with the reported NLPD distance, and interestingly, sensitivity is also bigger for more probable images, $\log(p(\mathbf{x}))$ via PixelCNN++.

but note that it is not crucial for the argument made here. In this regard, recent interesting models (Kingma & Dhariwal, 2018; Kadkhodaie et al., 2023) could also be used as probability estimates. Fig. 1 shows that cluttered images are successfully identified as less probable than smooth images by PixelCNN++, consistently with classical results on the average spectrum of natural images (Clarke, 1981; Atick et al., 1992; Malo et al., 2000; Simoncelli & Olshausen, 2001).

### 2.6 DISTORTED IMAGES

There are two factors to balance when looking for distortions so that the definition of *perceptual sensitivity* is meaningful. First; in order to understand the ratio in Eq. 1 as a variation *per unit of euclidean distortion*, we need the distorted image $\tilde{\mathbf{x}}$ to be close to the original $\mathbf{x}$. Secondly, the perceptual distances are optimised to recreate human judgements, so if the distortion is too small, such that humans are not able to see it, the distances are not trustworthy.

As such, we propose to use additive uniform noise on the surface of a sphere of radius $\epsilon$ around $\mathbf{x}$. We choose $\epsilon$ in order to generate $\tilde{\mathbf{x}}$ where the noise is small but still visible for humans ($\epsilon = 1$ in $32 \times 32 \times 3$ images with colour values in the range [0,1]). Certain radius (or Euclidean distance) corresponds to a unique RMSE, with $\epsilon = 1$ RMSE $= 0.018$ (about $2\%$ of the colour range). Examples of the noisy images can be seen in Fig. 1. Note that in order to use the perceptual distances and PixelCNN++, the image values must be in a specific range. After adding noise we clip the signal to keep that range. Clipping may modify substantially the Euclidean distance (or RMSE) so we only keep the images where RMSE $\in [0.017, 0.018]$, resulting in 48,046 valid images out of 50,000. For each image, we sample 10 noisy images $\tilde{\mathbf{x}}$. Probabilities estimated using PixelCNN++ around this amount of noise are smooth as seen in Appendix E.

## 3 EXPERIMENTS

Firstly, we empirically show the relation between the sensitivity and the candidate probabilistic factors using conditional histograms, and use information theory measures to tell which factors are most related to the sensitivity. Then, we explore polynomial combinations of several of these factors using regression models. Lastly, we restrict ourselves to the two most important factors and identify simple functional forms to predict perceptual sensitivity.

### 3.1 HIGHLIGHTING THE RELATION: CONDITIONAL HISTOGRAMS

Conditional histograms in Fig. 2 illustrate the relations between the sensitivity of each metric (Eq .1) conditioned to different probabilistic factors using the dataset described in Sec.2.6. In this case, the histograms describe the probabilities $\mathcal{P}(S \in [b_{j-1}, b_j]|X = x)$ where $S$ is certain perceptual sensitivity partitioned into $m = 30$ bins $[b_{j-1}, b_j)$, and $X$ is one of the possible probabilistic factors. This allows us to visually inspect which of the probabilistic factors are important for predicting sensitivity. The probabilistic factor used is given in each subplot title alongside with the Spearman correlation between perceptual sensitivity and the factor.

For all perceptual distances, $\log(p(\mathbf{x}))$ has a high correlation, and mostly follows a similar conditional distribution. NLPD is the only distance that significantly differs, with more sensitivity in mid-probability images. We also see a consistent increase in correlation and conditional means when looking at $\log(p(\tilde{\mathbf{x}}))$, meaning the log-likelihood of the noisy sample is more indicative of perceptual sensitivity for all tested distances. Also note that the standard deviation $\sigma(\mathbf{x})$ also has a strong (negative) correlation across the traditional distances, falling slightly with the deep learning based approaches. This is likely due to the standard deviation being closely related to the contrast of an image (Peli, 1990), and the masking effect: the sensitivity in known to decrease for higher contrasts (Legge & Foley, 1980; Martinez et al., 2019). Note that, as anticipated above, in Eq. 1, the integration of the sensitivity along a direction is related to the transduction of the visual system along that specific direction. With this in mind, the trends in the 2nd column or the last column in Fig. 2 are consistent with high slope in the transduction function for low contrast images (of high probability). Note, also that measures that take into account both points, $\int_{\mathbf{x}}^{\tilde{\mathbf{x}}} \log(p(\mathbf{x}))d\mathbf{x}$ and $\overrightarrow{J}_{\tilde{x}}(\mathbf{x})$, have lower correlation than just taking into account the noisy sample, with the latter being insignificant in predicting perceptual sensitivity.

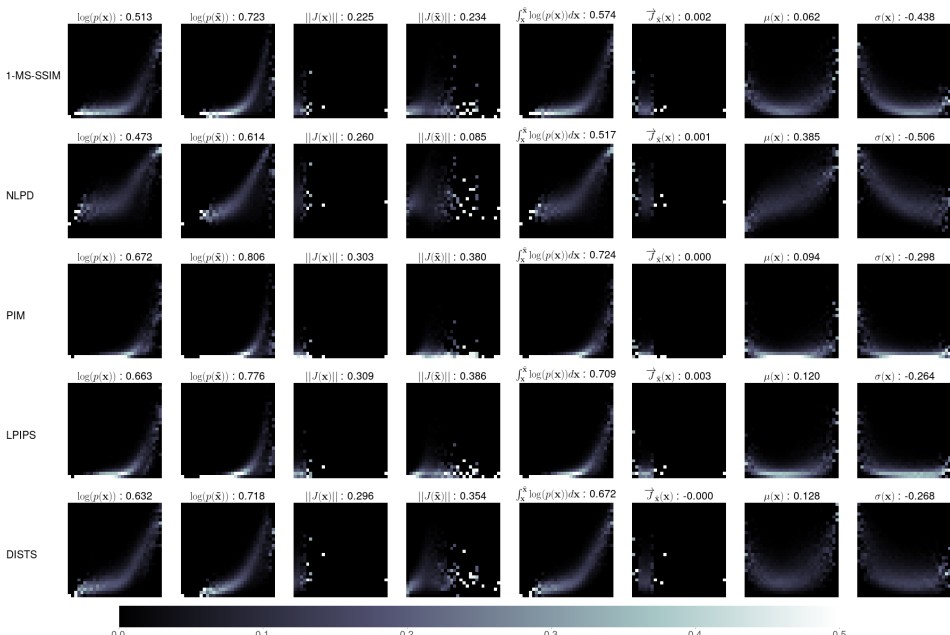

Figure 2: Conditional histograms for the dataset described in Sec.2.6. x-axis is the probability factor and y-axis is the sensitivity (Eq. 1) of the metric per row. Spearman correlation for each combination is in the title.

## 3.2 QUANTIFYING THE RELATION: MUTUAL INFORMATION

To quantify the ability of probabilistic factors to predict perceptual sensitivity, we use information theoretic measures. Firstly, we use mutual information which avoids the definition of a particular functional model and functional forms of the features. This analysis will give us insights into which of the factors derived from the statistics of the data can be useful in order to later define a functional model that relates statistics and perception.

The mutual information has been computed using all 48046 samples and using the Rotation Based Iterative Gaussianisation (RBIG) (Laparra et al., 2011) as detailed here (Laparra et al., 2020). Instead of the mutual information value ($I(X, Y)$), we report the Information coefficient of correlation (Linfoot, 1957) (ICC) since the interpretation is similar to the Pearson coefficient and allows for easy comparison, where $\text{ICC}(X, Y) = \sqrt{1 - e^{-2I(X,Y)}}$.

Fig. 3 shows the ICC between each isolated probabilistic factor and the sensitivity of different metrics. It is clear that the most important factor to take into account in most models (second in MS-SSIM) is the probability of the noisy image $\log(p(\tilde{\mathbf{x}}))$, a consistent result with the conditional histograms. Once we select the $\log(p(\tilde{\mathbf{x}}))$ as the most important factor, we explore which other factor should be included as the second term. In order to do so we analyse the ICC between each possible combination of two factors with each metric (right panel of Fig. 3). The matrix of possible pairs is depicted in detail for MS-SSIM (with factors in the axes and a colourbar for ICC), and corresponding matrices for the other metrics are also shown in smaller size. We can see that the general trend in four out of five metrics is the same: the pairs where $\log(p(\tilde{\mathbf{x}}))$ is involved have the maximum mutual information. The exception is MS-SSIM, which also behaves differently when using a single factor. In all the other cases the maximum relation with sensitivity is achieved when combining $\log(p(\tilde{\mathbf{x}}))$ with the standard deviation of the original image $\sigma(\mathbf{x})$.

The set of factors found to have a high relation with the variable to be predicted was expanded (one at a time) with the remaining factors. And we computed the mutual information and ICC for such expanded sets of factors. Reasoning as above (Fig. 3, right), we found that the next in relevance was $\log(p(\mathbf{x}))$, then the mean $\mu(\mathbf{x})$, then the gradient $||J(\mathbf{x})||$, to finally take into account *all* the probabilistic factors. Table 2 summarises the increase of ICC (the increase if information about the sensitivity) as we increase the number of probabilistic factors taken into account. In particular, we see that just by using $\log(p(\tilde{\mathbf{x}}))$ we can capture between $[0.55 - 0.71]$ of the information depending

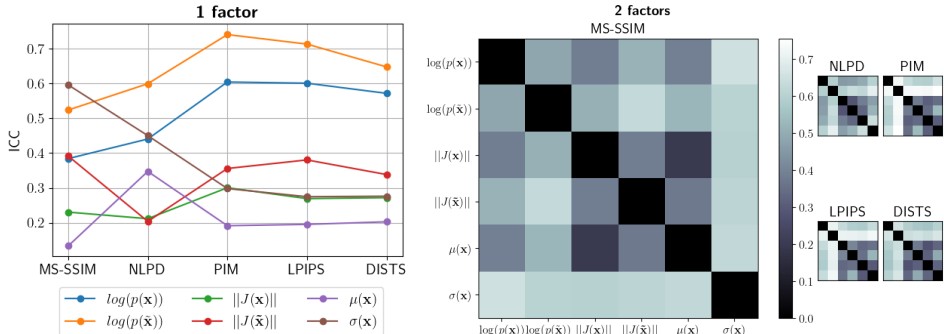

Figure 3: Information Coefficient of Correlation (ICC) between the sensitivity of different perceptual distances and isolated probabilistic factors (left: 1 factor) and different pairs of probabilistic factors (right: 2 factors). The considered factors are explicitly listed at the axes of the matrix of pairs for the MS-SSIM. The meaning of the entries for the matrices of the other distances is the same and all share the same colourbar for ICC.

on the metric. Then, for all the metrics, the consideration of additional factors improves ICC, but this improvement saturates as new factors contain less independent information about the sensitivity.

### 3.3 ACCURATE BUT NON-INTERPRETABLE MODEL: NON-PARAMETRIC REGRESSION

The goal is to achieve an interpretable straightforward parametric function predicting sensitivity based on probabilistic factors. Non-parametric models, while less interpretable, provide a performance reference (upper bound) for assessing the quality of our developed interpretable models. To do so, we consider a random forest regressor (Breiman, 2001) to predict the perceptual sensitivity from the probabilistic factors and 2nd order polynomial combinations of the factors, including the inverse probability for both original and noisy images. In regression trees is easy to analyse the relevance of each feature and compare between models trained on different perceptual sensitivities. We use a held out test set of 30% dataset in order to calculate correlations between predicted and ground truth. The average Pearson correlation obtained is 0.85, which serves as an illustrative upper bound reference for the interpretable functions proposed below in Sec. 3.4. We also trained a simple 3 layer multilayer perceptron on the same data and also achieved a correlation of 0.85.

Fig. 5 in Appendix B shows the 6 probabilistic factors with the larger importance across perceptual distances and their relative relevance for each distance. The Pearson (Spearman) correlations indicate how good the regression models are at predicting sensitivity. $\log(p(\tilde{\mathbf{x}}))$ is the most important factor, which agrees with the information theoretic analysis (Sec. 3.2). It has been suggested that the derivative of the log-likelihood is important in learning representations (Bengio et al., 2013) since modifications in the slope of the distribution imply label change and the score-matching objective makes use of the gradient. However, we find that this derivative has low mutual information with human sensitivity and low influence in the regression.

### 3.4 INTERPRETABLE MODEL: SIMPLE FUNCTIONAL FORMS

In order to get simple interpretable models of sensitivity, in this section we restrict ourselves to linear combinations of power functions of the probabilistic factors. We explore two situations: a

Table 2: Information Coefficient of Correlation between the sensitivity and groups of probabilistic factors.

| Factors | MS-SSIM | NLPD | PIM | LPIPS | DISTS | mean |
|---|---|---|---|---|---|---|
| 1D: $\log(p(\tilde{x}))$ | 0.55 | 0.57 | 0.71 | 0.68 | 0.61 | 0.62 |
| 2D: $\{\log(p(\tilde{x})), \sigma(x)\}$ | 0.57 | 0.66 | 0.72 | 0.69 | 0.63 | 0.65 |
| 3D: $\{\log(p(\tilde{x})), \sigma(x), \log(p(\mathbf{x}))\}$ | 0.68 | 0.68 | 0.72 | 0.69 | 0.65 | 0.68 |
| 4D: $\{\log(p(\tilde{x})), \sigma(x), \log(p(\mathbf{x})), \mu(x)\}$ | 0.68 | 0.76 | 0.75 | 0.71 | 0.66 | 0.71 |
| 5D: $\{\log(p(\tilde{\mathbf{x}})), \sigma(x), \log(p(\mathbf{x})), \mu(x), ||J(x)||\}$ | 0.68 | 0.78 | 0.75 | 0.73 | 0.66 | 0.72 |
| 6D: all factors | 0.71 | 0.79 | 0.76 | 0.73 | 0.66 | 0.73 |

single-factor model (1F), and a two-factor model (2F). According to the previous results on the ICC between the factors and sensitivity, the 1F model has to be based on $\log(p(\tilde{\mathbf{x}}))$, and the 2F model has to be based on $\log(p(\tilde{\mathbf{x}}))$ and $\sigma(\mathbf{x})$.

**1-factor model (1F).** In the literature, the use of $\log(p(\tilde{\mathbf{x}}))$ has been proposed using different exponents (see Table 1). Therefore, first, we analysed which exponents get better results for a single component, i.e. $S(\mathbf{x}, \tilde{\mathbf{x}}) = w_0 + w_1 \left(\log(p(\tilde{\mathbf{x}}))\right)^\gamma$. We found out that there is not a big difference between different exponents. Therefore we decided to explore the simplest solution: a regular polynomial (with natural numbers as exponents). We found that going beyond degree 2 does not substantially improve the correlation (detailed results are in Table 4 in Appendix C). We also explore a special situation where factors with different fractional and non-fractional exponents are linearly combined. The best overall reproduction of sensitivity across all perceptual metrics (Pearson correlation 0.73) can be obtained with a simple polynomial of degree two:

$$S(\mathbf{x}, \tilde{\mathbf{x}}) = w_0 + w_1 \, \log(p(\tilde{\mathbf{x}})) + w_2 \, \left(\log(p(\tilde{\mathbf{x}}))\right)^2, \tag{3}$$

where the values for the weights for each perceptual distance are in Appendix D (Table 6).

**2-factor model (2F).** In order to restrict the (otherwise intractable) exploration, we focus on polynomials that include the simpler versions of these factors, i.e. $\{\log(p(\tilde{\mathbf{x}})), (\log(p(\tilde{\mathbf{x}})))^2\}$ and $\{\sigma(\mathbf{x}), \sigma(\mathbf{x})^2, \sigma(\mathbf{x})^{-1}\}$, and the simplest products and quotients using both. We perform an ablation search of models that include these combinations as well as a LASSO regression (Tibshirani, 1996) (details in Appendix C (Table 5)). In conclusion, a model that combines good predictions and simplicity is the one that simply adds the $\sigma(\mathbf{x})$ factor to the 1-factor model:

$$S(\mathbf{x}, \tilde{\mathbf{x}}) = w_0 + w_1 \, \log(p(\tilde{\mathbf{x}})) + w_2 \, \left(\log(p(\tilde{\mathbf{x}}))\right)^2 + w_3 \, \sigma(\mathbf{x}) \tag{4}$$

where the values of the weights for the different metrics are in Appendix D (Table 7). The 2-factor model obtains an average Pearson correlation of 0.79 across the metrics, which implies an increase of 0.06 in correlation in regard to the 1-factor model. As a reference, a model that includes all nine analysed combinations for the two factors achieves an average 0.81 correlation (not far from the performance obtained with the non-parametric regressor, 0.85, in Sec. 3.3).

**Generic coefficients.** The specific coefficients for all the metrics are very similar. Therefore we normalise each set of coefficients with respect to $w_0$, and use the mean weights $\{w_0, w_1, w_2, w_3\}$ as a generic set of coefficients. It achieves very good correlation, 1F: $\rho = 0.74 \pm 0.02$; 2F: $\rho = 0.77 \pm 0.02$ (details are in the Appendix D).

## 4 VALIDATION: REPRODUCTION OF CLASSICAL PSYCHOPHYSICS

Variations of human visual sensitivity depending on luminance, contrast, and spatial frequency were identified by classical psychophysics (Weber, 1846; Legge & Foley, 1980; Campbell & Robson, 1968; Georgeson & Sullivan, 1975) way before the advent of the state-of-the-art perceptual distances considered here. Therefore, the reproduction of the trends of those classic behaviours is an independent way to validate the proposed models since it is not related to the recent image quality databases which are at the core of the perceptual distances.

We perform two experiments for this validation: (1) we compute the sensitivity to sinusoidal gratings of different frequencies but the same energy (or contrast), at different contrast levels, and (2) we compute the sensitivity for natural images depending on mean luminance and contrast. The first experiment is related to the concept of Contrast Sensitivity Function (CSF) (Campbell & Robson, 1968): the human transfer function in the Fourier domain, just valid for low contrasts, and the decay of such filter with contrast, as shown in the suprathreshold contrast matching experiments Georgeson & Sullivan (1975). The second experiment is related to the known saturation of the response (reduction of sensitivity) for higher luminances: the Weber law (Weber, 1846), and its equivalent (also reduction of the sensitivity) for contrast due to masking (Legge & Foley, 1980).

Gratings in Experiment 1 were generated with a fixed average luminance of 40 $cd/m^2$, and frequencies in the range [1,8] cycles per degree (cpd). We considered three Michelson contrast levels $c = \{0.2, 0.4, 0.7\}$. Barely visible noise in $\epsilon = 1$ sphere was added to the gratings to generate the distorted stimuli $\tilde{\mathbf{x}}$. In Experiment 2, we define contrast as $c = \sigma(\mathbf{x})\sqrt{2}/\mu(\mathbf{x})$, which is applicable to natural images and reduces to Michelson contrast for gratings (Peli, 1990). We edited images

from CIFAR10 to have the same contrast levels as in Experiment 1 and we explored the range of luminances $[60, 140]cd/m^2$ so that the digital values still were in the [0,255] range. We then averaged the sensitivities for 1000 images in CIFAR-10. Averaged sensitivities computed using the *generic coefficients* over 100 noise samples for both experiments are plotted in Fig. 4, and illustrative test images are shown in Fig. 7 in Appendix F.

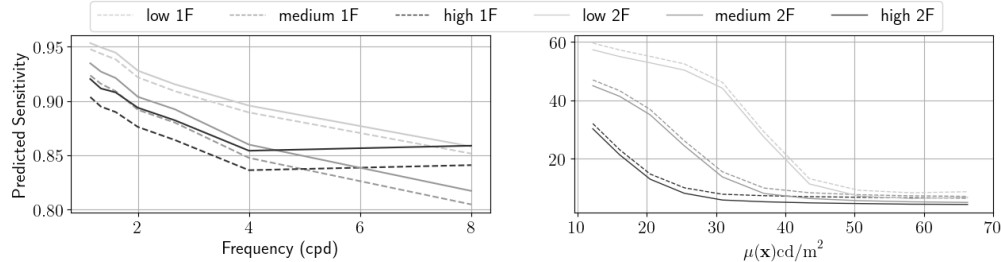

Figure 4: Validation experiment 1 (Contrast Sensitivity Function, *left*), and validation experiment 2 (Weber law and masking, *right*) for the 1-factor (1F) and 2-factors (2F) using the generic coefficients. Color corresponds to the different contrast levels (low=0.2, medium=0.4, high=0.7).

Regarding the sensitivity to gratings (left), results reproduce the decay of the CSF at high frequencies, its decay with contrast, and how the shape flattens with contrast, consistently with (Campbell & Robson, 1968; Georgeson & Sullivan, 1975). For sensitivity at different luminances and contrasts (right) there is a simultaneous reduction in both, consistent with the literature (Weber, 1846; Legge & Foley, 1980). Interestingly, both models 1-F and 2-F follow the same trends.

## 5 DISCUSSION AND CONCLUSION

We show that several functions of image probability computed from recent generative models share substantial information with the sensitivity of state-of-the-art perceptual metrics (mean ICC of 0.73, Sec. 3.2). Alongside this, a Random Forest regressor that predicts sensitivity from polynomial combinations of these probabilistic factors obtains an average Pearson correlation of 0.85 with actual sensitivity (Sec. 3.3). According to the shared information, the factors can be ranked as: $\{\log(p(\tilde{\mathbf{x}})), \sigma(\mathbf{x}), \log(p(\mathbf{x})), \mu(\mathbf{x}), ||J(\mathbf{x})||\}$, in agreement with the relevance given by the regression tree that uses the same features. Using only $\log(p(\tilde{\mathbf{x}}))$ obtains an average ICC of 0.62, and a simple quadratic model including only this factor achieves a 0.73 Pearson correlation. This is due to simultaneously including information about the distribution of the original image (in a differential context $p(\tilde{\mathbf{x}}) \approx p(\mathbf{x})$), and also about the specific direction of distortion. After an exhaustive ablation study keeping the factors with the highest ICC values, we propose simple functional forms of the sensitivity-probability relation using just 1 or 2 factors (Eqs. 3 based on $\log(p(\tilde{\mathbf{x}}))$, and Eq. 4 based on $\log(p(\tilde{\mathbf{x}}))$ and $\sigma(\mathbf{x})$), that obtain Pearson correlations with the metric sensitivities of $\rho = 0.74 \pm 0.02$; 2F: $\rho = 0.77 \pm 0.02$. These simple functions of image probability are validated through the reproduction of trends of the human frequency sensitivity, its suprathreshold variation, the Weber law, and contrast masking (Sec. 4).

This study inherits the limitations of the PixelCNN++ probability model, trained on a limited set of small images with restricted contrast, luminance, color, and content range. To ensure reliable $p(\mathbf{x})$, we confined ourselves to a similar set of images. Additionally, we employ perceptual metrics as a surrogate for human perception. Despite correlations with humans, even state-of-the-art metrics have limitations. Consequently, further psychophysical work is essential to validate relationships with $p(\mathbf{x})$. While the use of proxies is justified due to the impracticality of gathering sufficient psychophysical data, the insights gained can guide experimental efforts towards specific directions.

Nevertheless, shared information and correlations are surprisingly high given that the statistics of the environment is not the only in driving factor in vision (Erichsen & Woodhouse, 2012; Sterling & Laughlin, 2015). In fact, this successful prediction of sensitivity from accurate measures of image probability with such simple functional forms is a renewed *direct* evidence of the relevance of Barlow Hypothesis. Moreover, our results are in contrast with intuitions about the importance of the distribution gradient (Bengio et al., 2013): we found that the gradient of the probability has low shared mutual information and low feature influence in the human sensitivity.

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
