## A PERFORMANCE OF THE CONSIDERED PERCEPTUAL METRICS

Image quality metrics are able to predict visual human perception up to some level. In Table 3 we show results of the different image quality metrics used in this work (see sec. 2.4) when evaluated on human rates databases. Although the metrics are not perfect, it is clear that they are a good proxy of human perception. We show results on a traditional perceptual dataset using large images, TID2013 (Ponomarenko et al., 2013), and a more recent dataset with smaller images (size $64 \times 64$) but more distortions as using neural networks based ones, such as super-resolution, BAPPS (Zhang et al., 2018).

Table 3: Pearson and Spearman correlations with human opinion in TID2013, and agreement with human judgement (in %) in BAPPS.

|  | MSSSIM | NLPD | PIM | LPIPS | DISTS |
|---|---|---|---|---|---|
| TID2013 $\rho_p$ ($\rho_s$) | 0.78 (0.80) | 0.84 (0.80) | 0.62 (0.65) | 0.74 (0.67) | 0.86 (0.83) |
| BAPPS (%) | 61.7 | 61.5 | 64.5 | 69.2 | 69.0 |

## B RELEVANCE OF FACTORS USING RANDOM FOREST REGRESSORS

A random forest regression was fit using multiple combinations of the probability-related factors (see sec. 2.3) to predict the sensitivity of the different image quality metrics used in this work (see sec. 2.4). A total of 55 combinations were introduced as inputs. Values of feature importance are normalised so that they sum to 1 for each model for easy comparison. Fig. 5 shows the most relevant ones selected by the random forest algorithm.

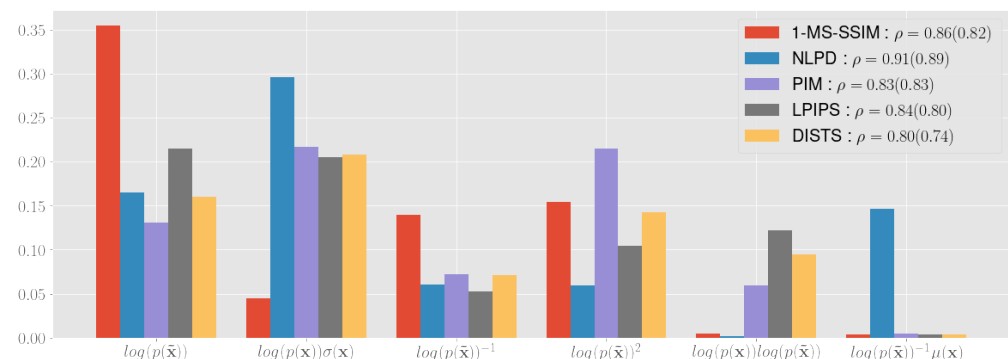

Figure 5: Top 6 feature importance from Random Forest regressors trained on polynomial combinations of the probabilistic factors in order to predict perceptual sensitivity. A separate model was trained for each perceptual distance. In the legend we include the Pearson (Spearman) correlation between the predictions and ground truth for a held out test set 30% of the dataset.

## C PARAMETER SELECTION FOR THE FUNCTIONAL FORMS

Here we show the details for the selection of the parameters in Sec. 4. For the one-factor model, we tried different possibilities on the selected factor $\log(p(\tilde{\mathbf{x}}))$, details on the correlation of the different possibilities with sensitivity of perceptual measures are given in Table 4 and Sec 3.4.

For the two-factors model, we took the ones obtained in Sec 4.1 (i.e. $b$ (bias), $\log(p(\tilde{\mathbf{x}}))$, and $(\log(p(\tilde{\mathbf{x}})))^2$), and factors of the standard deviation $\sigma_x$ as suggested by the mutual information in Sec. 3.2. The combinations of the standard deviation have been alone: $\sigma_x$, $\frac{1}{\sigma_x}$, $\sigma_x^2$, and combined with the probability of the noisy image: $\frac{\log(p(\tilde{\mathbf{x}}))}{\sigma_x}$, $\frac{\sigma_x}{\log(p(\tilde{\mathbf{x}}))}$, and $\log(p(\tilde{\mathbf{x}}))\sigma_x$.

There are 9 candidates but we want the most compact and interpretable model. Analysing all the possible combinations is intractable so we are going to perform an ablation study: we are going

Table 4: Pearson correlation obtained between the prediction of the model and the sensitivity for different IQMs. All models are designed using versions of $\log(p(\tilde{\mathbf{x}}))$ as input factor. The models are simple models with only *one coefficient*, $S(\mathbf{x}, \tilde{\mathbf{x}}) = w_0 + w_1 \left(\log(p(\tilde{\mathbf{x}}))\right)^\gamma$, or *polynomials* of different degrees (d). The model *Frac\** is a special polynomial with exponents: $[0.3, 0.2, 0.1, 0, 1, 2, 3]$.

|  | MSSIM | NLPD | PIM | LPIPS | DISTS | Mean |
|---|---|---|---|---|---|---|
| One coefficient | | | | | | |
| $\gamma = 1/10$ | 0.71 | 0.64 | 0.66 | 0.69 | 0.72 | **0.68** |
| $\gamma = 1/5$ | 0.71 | 0.64 | 0.66 | 0.69 | 0.72 | **0.68** |
| $\gamma = 1/3$ | 0.71 | 0.64 | 0.66 | 0.69 | 0.72 | **0.68** |
| $\gamma = 1/2$ | 0.71 | 0.63 | 0.66 | 0.69 | 0.72 | **0.68** |
| $\gamma = 2$ | 0.69 | 0.63 | 0.64 | 0.68 | 0.71 | **0.67** |
| $\gamma = -1$ | 0.72 | 0.65 | 0.66 | 0.71 | 0.73 | **0.69** |
| $\gamma = 1$ | 0.7 | 0.63 | 0.65 | 0.68 | 0.72 | **0.68** |
| Polynomials | | | | | | |
| d = 2 | 0.76 | 0.65 | 0.75 | 0.76 | 0.74 | **0.73** |
| d = 3 | 0.76 | 0.65 | 0.75 | 0.75 | 0.74 | **0.73** |
| d = 6 | 0.75 | 0.64 | 0.73 | 0.74 | 0.74 | **0.72** |
| *Frac\** | 0.76 | 0.65 | 0.76 | 0.76 | 0.74 | **0.73** |

to discard different candidates sequentially starting from the largest model (9 candidates). Besides, we are going to use LASSO regression with different amounts of regularisation to get models with different amounts of factors as comparison to the ablation study. Results in Table 5 are shown in descending number of factors used. For each step, we remove the factor (or factors) that less influence has in the correlation. Besides, we show the correlation given by LASSO models where the regularization parameter has been adjusted in order to have the same number of factors. A model with 6 factors (number 17) has the same correlation (0.81) as the one with all the factors (number 1). The best trade-off between the number of factors and correlation is for models 25, 26, and 27, with 4 factors and a correlation of 0.79. We chose as our final functional model in Sec. 4.2 the model 25 as its factors involve less computations.

Table 5: Linear models using different combinations of $b$ (bias), $log(p(\tilde{x})) = p$ and $\sigma(x) = s$, where 1 indicates the factor is included in the model, and 0 it is ablated. Models with $*$ in the model number (# M) have been computed using Lasso and a corresponding regularization parameter in order to have a particular number of combinations (# C). For the perceptual metrics, the Pearson correlation between the predicted and ground truth on a test set of CIFAR10 is reported.

| $b$ | $p$ | $p^2$ | $s$ | $s^2$ | $\frac{1}{s}$ | $\frac{p}{s}$ | $\frac{s}{p}$ | $ps$ | MSSIM | NLPD | PIM | LPIPS | DISTS | Mean | # C | #M |
|---|---|---|---|---|---|---|---|---|---|---|---|---|---|---|---|---|
| 1 | 1 | 1 | 1 | 1 | 1 | 1 | 1 | 1 | 0.85 | 0.78 | 0.81 | 0.81 | 0.78 | **0.806** | 9 | 1 |
| | | | | | | | | | **8 combinations** | | | | | | | |
| 1 | 1 | 1 | 0 | 1 | 1 | 1 | 1 | 1 | 0.85 | 0.78 | 0.8 | 0.8 | 0.78 | **0.802** | 8 | 2 |
| 1 | 1 | 1 | 1 | 0 | 1 | 1 | 1 | 1 | 0.85 | 0.78 | 0.81 | 0.81 | 0.78 | **0.806** | 8 | 3 |
| 1 | 1 | 1 | 1 | 1 | 0 | 1 | 1 | 1 | 0.85 | 0.78 | 0.81 | 0.81 | 0.78 | **0.806** | 8 | 4 |
| 1 | 1 | 1 | 1 | 1 | 1 | 0 | 1 | 1 | 0.85 | 0.78 | 0.81 | 0.81 | 0.78 | **0.806** | 8 | 5 |
| 1 | 1 | 1 | 1 | 1 | 1 | 1 | 0 | 1 | 0.85 | 0.78 | 0.8 | 0.8 | 0.78 | **0.802** | 8 | 6 |
| 1 | 1 | 1 | 1 | 1 | 1 | 1 | 1 | 0 | 0.85 | 0.78 | 0.8 | 0.8 | 0.78 | **0.802** | 8 | 7 |
| | | | | | | | | | **7 combinations** | | | | | | | |
| 0 | 1 | 1 | 1 | 1 | 1 | 1 | 1 | 0 | 0.83 | 0.75 | 0.77 | 0.77 | 0.77 | **0.778** | 7 | 8 |
| 1 | 0 | 1 | 1 | 1 | 1 | 1 | 1 | 0 | 0.82 | 0.78 | 0.77 | 0.78 | 0.77 | **0.784** | 7 | 9 |
| 1 | 1 | 0 | 1 | 1 | 1 | 1 | 1 | 0 | 0.82 | 0.78 | 0.76 | 0.77 | 0.76 | **0.778** | 7 | 10 |
| 1 | 1 | 1 | 0 | 1 | 1 | 1 | 1 | 0 | 0.84 | 0.78 | 0.8 | 0.8 | 0.78 | **0.8** | 7 | 11 |
| 1 | 1 | 1 | 1 | 0 | 1 | 1 | 1 | 0 | 0.85 | 0.78 | 0.8 | 0.8 | 0.78 | **0.802** | 7 | 12 |
| 1 | 1 | 1 | 1 | 1 | 0 | 1 | 1 | 0 | 0.85 | 0.78 | 0.8 | 0.8 | 0.78 | **0.802** | 7 | 13 |
| 1 | 1 | 1 | 1 | 1 | 1 | 0 | 1 | 0 | 0.85 | 0.78 | 0.8 | 0.8 | 0.78 | **0.802** | 7 | 14 |
| 1 | 1 | 1 | 1 | 1 | 1 | 1 | 0 | 0 | 0.84 | 0.78 | 0.8 | 0.8 | 0.78 | **0.8** | 7 | 15 |
| 0 | 1 | 1 | 1 | 1 | 1 | 1 | 0 | 1 | 0.84 | 0.79 | 0.78 | 0.8 | 0.78 | **0.798** | 7 | 16* |
| | | | | | | | | | **6 combinations** | | | | | | | |
| 1 | 1 | 1 | 1 | 0 | 0 | 0 | 1 | 1 | 0.85 | 0.78 | 0.81 | 0.81 | 0.78 | **0.806** | 6 | 17 |
| 0 | 1 | 1 | 0 | 1 | 1 | 1 | 0 | 1 | 0.84 | 0.79 | 0.78 | 0.8 | 0.78 | **0.798** | 6 | 18* |
| | | | | | | | | | **5 combinations** | | | | | | | |
| 1 | 0 | 1 | 1 | 0 | 0 | 0 | 1 | 1 | 0.84 | 0.78 | 0.78 | 0.79 | 0.77 | **0.792** | 5 | 19 |
| 1 | 1 | 0 | 1 | 0 | 0 | 0 | 1 | 1 | 0.84 | 0.78 | 0.78 | 0.79 | 0.77 | **0.792** | 5 | 20 |
| 1 | 1 | 1 | 0 | 0 | 0 | 0 | 1 | 1 | 0.84 | 0.78 | 0.8 | 0.8 | 0.78 | **0.8** | 5 | 21 |
| 1 | 1 | 1 | 1 | 0 | 0 | 0 | 0 | 1 | 0.84 | 0.78 | 0.8 | 0.8 | 0.78 | **0.8** | 5 | 22 |
| 1 | 1 | 1 | 1 | 0 | 0 | 0 | 1 | 0 | 0.84 | 0.78 | 0.8 | 0.8 | 0.78 | **0.8** | 5 | 23 |
| 0 | 1 | 1 | 0 | 0 | 1 | 1 | 0 | 1 | 0.83 | 0.79 | 0.78 | 0.8 | 0.78 | **0.796** | 5 | 24* |
| | | | | | | | | | **4 combinations** | | | | | | | |
| 1 | 1 | 1 | 1 | 0 | 0 | 0 | 0 | 0 | 0.83 | 0.78 | 0.77 | 0.78 | 0.77 | **0.786** | 4 | 25 |
| 1 | 1 | 1 | 0 | 0 | 0 | 0 | 1 | 0 | 0.83 | 0.78 | 0.78 | 0.78 | 0.77 | **0.788** | 4 | 26 |
| 1 | 1 | 1 | 0 | 0 | 0 | 0 | 0 | 1 | 0.83 | 0.78 | 0.77 | 0.78 | 0.77 | **0.786** | 4 | 27 |
| 1 | 1 | 0 | 0 | 0 | 0 | 0 | 1 | 1 | 0.8 | 0.78 | 0.72 | 0.74 | 0.76 | **0.76** | 4 | 28 |
| 1 | 0 | 1 | 0 | 0 | 0 | 0 | 1 | 1 | 0.79 | 0.77 | 0.7 | 0.73 | 0.75 | **0.748** | 4 | 29 |
| 0 | 1 | 1 | 1 | 0 | 0 | 0 | 1 | 0 | 0.76 | 0.65 | 0.75 | 0.76 | 0.74 | **0.732** | 4 | 30 |
| 1 | 0 | 1 | 1 | 0 | 0 | 0 | 1 | 0 | 0.79 | 0.77 | 0.7 | 0.72 | 0.75 | **0.746** | 4 | 31 |
| 1 | 1 | 0 | 1 | 0 | 0 | 0 | 1 | 0 | 0.8 | 0.77 | 0.71 | 0.74 | 0.75 | **0.754** | 4 | 32 |
| 0 | 1 | 1 | 0 | 0 | 0 | 1 | 0 | 1 | 0.83 | 0.79 | 0.78 | 0.76 | 0.77 | **0.786** | 4 | 33* |
| | | | | | | | | | **3 combinations** | | | | | | | |
| 1 | 1 | 1 | 0 | 0 | 0 | 0 | 0 | 0 | 0.76 | 0.65 | 0.75 | 0.76 | 0.74 | **0.732** | 3 | 34 |
| 1 | 1 | 0 | 0 | 0 | 0 | 0 | 1 | 0 | 0.79 | 0.77 | 0.69 | 0.72 | 0.75 | **0.744** | 3 | 35 |
| 1 | 1 | 0 | 0 | 0 | 0 | 0 | 0 | 1 | 0.78 | 0.77 | 0.68 | 0.71 | 0.75 | **0.738** | 3 | 36 |
| 1 | 1 | 0 | 1 | 0 | 0 | 0 | 0 | 0 | 0.79 | 0.77 | 0.69 | 0.71 | 0.75 | **0.742** | 3 | 37 |
| 1 | 0 | 1 | 0 | 0 | 0 | 0 | 0 | 1 | 0.78 | 0.77 | 0.68 | 0.7 | 0.74 | **0.734** | 3 | 38 |
| 1 | 0 | 1 | 0 | 0 | 0 | 0 | 1 | 0 | 0.78 | 0.77 | 0.68 | 0.71 | 0.74 | **0.736** | 3 | 39 |
| 1 | 0 | 1 | 1 | 0 | 0 | 0 | 0 | 0 | 0.78 | 0.77 | 0.68 | 0.71 | 0.74 | **0.736** | 3 | 40 |
| 1 | 0 | 0 | 0 | 0 | 0 | 0 | 1 | 1 | 0.73 | 0.75 | 0.62 | 0.65 | 0.7 | **0.69** | 3 | 41 |
| 1 | 0 | 0 | 1 | 0 | 0 | 0 | 1 | 0 | 0.74 | 0.75 | 0.62 | 0.65 | 0.7 | **0.692** | 3 | 42 |
| 1 | 0 | 0 | 1 | 0 | 0 | 0 | 0 | 1 | 0.73 | 0.75 | 0.61 | 0.64 | 0.7 | **0.686** | 3 | 43 |
| 0 | 0 | 1 | 0 | 0 | 0 | 1 | 0 | 1 | 0.8 | 0.78 | 0.73 | 0.72 | 0.75 | **0.756** | 3 | 44* |
| | | | | | | | | | **2 combinations** | | | | | | | |
| 1 | 1 | 0 | 0 | 0 | 0 | 0 | 0 | 0 | 0.7 | 0.63 | 0.65 | 0.68 | 0.72 | **0.676** | 2 | 45 |
| 1 | 0 | 1 | 0 | 0 | 0 | 0 | 0 | 0 | 0.69 | 0.63 | 0.64 | 0.68 | 0.71 | **0.67** | 2 | 46 |
| 1 | 0 | 0 | 1 | 0 | 0 | 0 | 0 | 0 | 0.43 | 0.52 | 0.29 | 0.28 | 0.3 | **0.364** | 2 | 47 |
| 1 | 0 | 0 | 0 | 0 | 0 | 0 | 1 | 0 | 0.34 | 0.43 | 0.21 | 0.2 | 0.2 | **0.276** | 2 | 48 |
| 1 | 0 | 0 | 0 | 0 | 0 | 0 | 0 | 1 | 0.51 | 0.59 | 0.36 | 0.36 | 0.38 | **0.44** | 2 | 49 |
| 0 | 0 | 1 | 0 | 0 | 0 | 1 | 0 | 0 | 0.71 | 0.76 | 0.68 | 0.71 | 0.74 | **0.72** | 2 | 50* |

Table 6: Coefficients for the Eq. 3 for each metric, and each weight normalised by $w_0$ in order to compare between metrics.

| Coefs | MSSIM | NLPD | PIM | LPIPS | DISTS | Mean |
|---|---|---|---|---|---|---|
| $w_0$ | 29.5 | 65 | 15400 | 198 | 161 | |
| $w_1$ | $4.9 \times 10^{-3}$ | $9.5 \times 10^{-3}$ | 2.62 | $3.33 \times 10^{-2}$ | $2.58 \times 10^{-2}$ | |
| $w_2$ | $2.05 \times 10^{-7}$ | $3.62 \times 10^{-7}$ | $1.11 \times 10^{-4}$ | $1.41 \times 10^{-6}$ | $1.05 \times 10^{-6}$ | |
| $w_0/w_0$ | 1 | 1 | 1 | 1 | 1 | 1 |
| $w_1/w_0$ | $1.66 \times 10^{-4}$ | $1.46 \times 10^{-4}$ | $1.70 \times 10^{-4}$ | $1.68 \times 10^{-4}$ | $1.60 \times 10^{-4}$ | $1.62 \times 10^{-4}$ |
| $w_2/w_0$ | $6.94 \times 10^{-9}$ | $5.57 \times 10^{-9}$ | $7.21 \times 10^{-9}$ | $7.12 \times 10^{-9}$ | $6.52 - \times 10^{-9}$ | $6.67 \times 10^{-9}$ |

Table 7: Coefficients for the Eq. 4 for each metric, and each weight normalised by $w_0$ in order to compare between metrics.

| Coefs | MSSIM | NLPD | PIM | LPIPS | DISTS | Mean |
|---|---|---|---|---|---|---|
| $w_0$ | 28 | 58 | 15100 | 194 | 156 | |
| $w_1$ | $4.69 \times 10^{-3}$ | $8.19 \times 10^{-3}$ | 2.57 | $3.26 \times 10^{-2}$ | $2.49 \times 10^{-2}$ | |
| $w_2$ | $1.96 \times 10^{-7}$ | $3.09 \times 10^{-7}$ | $1.09 \times 10^{-4}$ | $1.37 \times 10^{-6}$ | $1.00 \times 10^{-6}$ | |
| $w_3$ | $-0.597$ | $-3.74$ | $-141$ | $-1.93$ | $-2.54$ | |
| $w_0/w_0$ | 1 | 1 | 1 | 1 | 1 | 1 |
| $w_1/w_0$ | $1.68 \times 10^{-4}$ | $1.41 \times 10^{-4}$ | $1.70 \times 10^{-4}$ | $1.68 \times 10^{-4}$ | $1.60 \times 10^{-4}$ | $1.61 \times 10^{-4}$ |
| $w_2/w_0$ | $7.00 \times 10^{-9}$ | $5.33 \times 10^{-9}$ | $7.22 \times 10^{-9}$ | $7.06 \times 10^{-9}$ | $6.41 \times 10^{-9}$ | $6.6 \times 10^{-9}$ |
| $w_3/w_0$ | $-0.021$ | $-0.064$ | $-0.0093$ | $-0.0010$ | $-0.016$ | $-0.024$ |

## D   COEFFICIENTS OF THE FUNCTIONAL FORMS

In Sec. 4 we propose Eqs. 3 and 4 as estimators of the perceptual sensitivity for 1- and 2-factor models respectively. In Tables 6 and 7 we give the actual weights obtained in the experiments for each distance.

Each metric has a different interpretation of the sensitivity units, therefore the weights (coefficients) of the proposed equations are different for each measure. However, note that the proportion of the coefficients for each probability factor are very similar. Actually by normalizing the coefficients by the *bias* ($b$) the normalized coefficients are very similar (see Tables 6 and 7).

By simply taking the mean of each normalized coefficient for the six metrics we get the last column which is a set of coefficients that can be use to predict the sensitivity for each metric (up to the normalizing factor which does not affecto to the correlatiobn). The Pearson correlations using the models with the *Mean* coefficients are shown in Table 8. These simple and general models get very good correlation with the sensitivity of all the metrics simultaneously.

As a summary, if one takes one of the models in eqs. 3 or 3, using the coefficients in the mean column in Tables 6 or 7 respectively, can get a good estimation of the sensitivity of a perceptual measure.

Table 8: Pearson correlations obtained using a single set of coefficients ("Mean" column in Tables 6 and 7) for the 1F and 2F models. Pearson for coefficients fitted for each particular metric are given for comparison (correlations are the same as in Table 4 row $d = 2$) for 1F, and Table 5 first row 4 combinations.

| | MSSIM | NLPD | PIM | LPIPS | DISTS |
|---|---|---|---|---|---|
| Mean coefficients 1F | 0.75 | 0.64 | 0.73 | 0.75 | 0.74 |
| Specific coefficients 1F | 0.76 | 0.65 | 0.75 | 0.76 | 0.74 |
| Mean coefficients 2F | 0.82 | 0.76 | 0.75 | 0.76 | 0.77 |
| Specific coefficients 2F | 0.83 | 0.78 | 0.77 | 0.78 | 0.77 |

## E  PIXELCNN++ SMOOTHNESS

While in the original PixelCNN++ paper authors talk about PDF as the magnitude estimated by the model Salimans et al. (2017), there are some parts of the model that perform a kind of quantisation. one can wonder if this quantisation can affect the experiments. In Fig. 6 we show the probability of an image with an increasing level of standard deviation for the noise, we kept the noise pattern constant (i.e. fixed seed) but we modify the magnitude of the noise. While at extremely low levels the effect of the quantisation is noticeable, at very low and low level of noise the estimations vary softly. As a reference we show the level of noise analysed in this work, it is really far from the quantisation region.

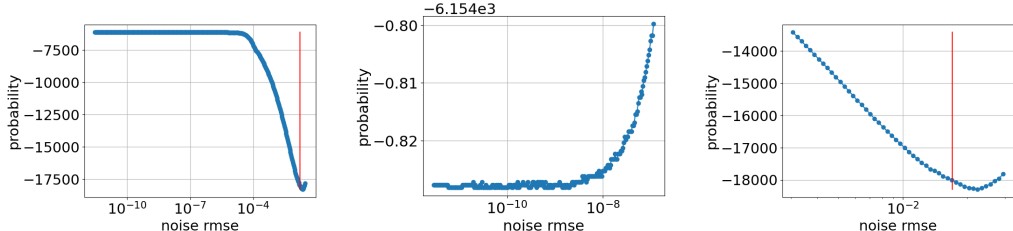

Figure 6: Probability estimations and the corresponding rmse between the original and the distorted image using uniform noise. We use a fixed image and the same noise pattern but we increase the magnitude of the noise, the red line marks the level of noise used in our experiments. Left: full plot, the red line is the noise we add throughout the paper. Middle: details at extremely low level of noise (quantisation effect is visible). Right: details at very low level of noise (quantisation effect is negligible).

## F  PERCEPTUAL TESTS ON THE MODEL

Here we show the stimuli used for both perceptual tests performed using the proposed models and the schematic for the perceptual experiments.

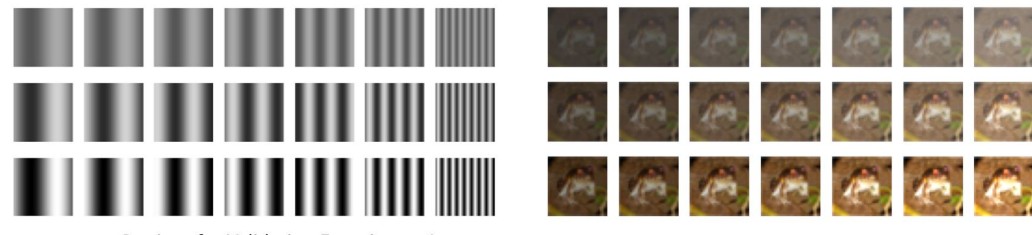

Figure 7: Stimuli for the perceptual tests of the proposed model. (left) Gratings are generated at wavelengths that can be accurately shown in a $32 \times 32$ spatial size, generated at contrasts of $[0.2, 0.4, 0.7]$. (right) Shows one example of editing an image from the CIFAR10 dataset to vary contrast of $[0.2, 0.4, 0.7]$ and luminance values in $[60, 140]$.

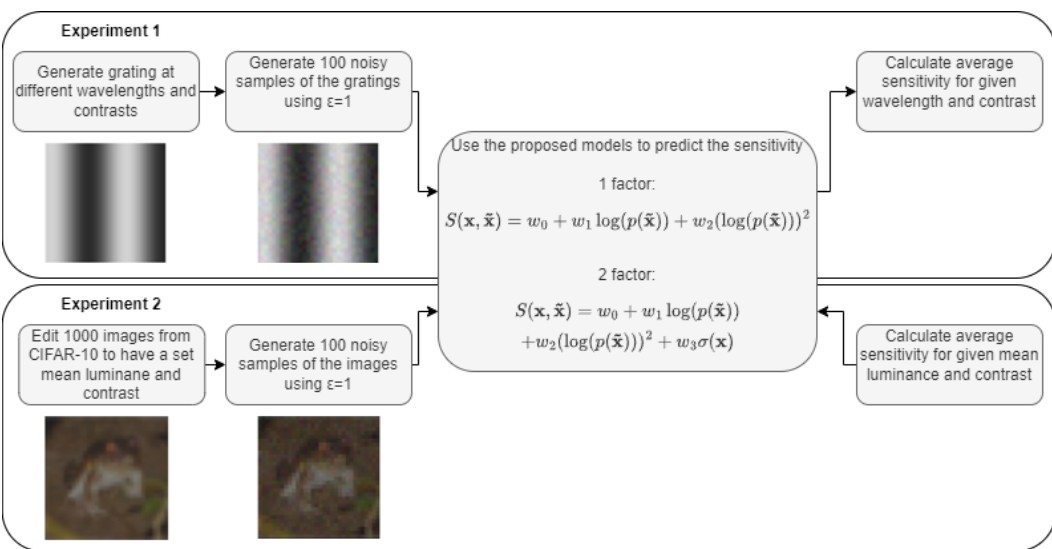

Figure 8: Schematic describing the perceptual experiments 1 and 2.