# OpenReview forum: "Disentangling the Link Between Image Statistics and Human Perception"
_ICLR.cc/2024/Conference — Submitted to ICLR 2024_

### Official Review · Reviewer_Cqjc · 2023-10-30

**Soundness:** 2 fair
**Presentation:** 3 good
**Contribution:** 2 fair
**Rating:** 5
**Confidence:** 5

**Summary:**

This paper investigates the use of probability-related factors to explain/predict human perception (approximated by sensitivity of
state-of-the-art image quality metrics).

**Strengths:**

1. The problem is of fundamental and theoretical importance.

2. The related work, especially Sec. 2.2 is a delightful read to the reviewer.

**Weaknesses:**

1. The definition of perceptual sensitivity in Eq. (1) is debatable.  For any given $D_p(x,\hat{x})$, it is easy to come up with $\hat{x}$ to be a counterexample of $S(x,\hat{x})$, i.e., big ratio corresponding to low human sensitivity and vice versa. For example, the synthesis of $\hat{x}$ can be performed by the maximum differentiation competition [Wang and Simoncelli] or the perceptual attack [Zhang et al.].

2. The authors should clearly state the meaning of $p(x)$: is it probability density function (PDF) or probability mass function (PMF)? Working with PDF is less reasonable, if the learned distribution is not smooth.

3. Perturbing images with additive uniform noise makes the results in this paper less interesting. Any ideas on the optimal perturbation for the same goal (using probability-related factors to explain/predict human perception)?

4. Parameteric prediction in Eq. (3) and Eq. (4) can be trivial and thus meaningless with a deliberately chosen set of $\hat{\mathcal{X}}_1$={$\{\hat{x}\}$}. Putting another way, it is not hard to come up with another  $\hat{\mathcal{X}}_2$={$\{\hat{x}\}$} to make the parametric prediction nearly impossible.

5. How to apply the computational analysis in the paper? Can the results reflect which quality metrics are better explaining human perception?

**Questions:**

1. The reviewer fails to understand the message of Fig. 2.

**Details Of Ethics Concerns:**

N.A.

---

> ### Author Response · Authors · 2023-11-16
> **Rebuttal for Reviewer Cqjc**
>
> We thank you for the review.
>
> > The definition of perceptual sensitivity in Eq. (1) is debatable. For any given $D_p(\mathbf{x}, \hat{\mathbf{x}})$, it is easy to come up with $\hat{\mathbf{x}}$ to be a counterexample of $S(\mathbf{x}, \hat{\mathbf{x}})$, i.e., big ratio corresponding to low human sensitivity and vice versa. For example, the synthesis of $\hat{\mathbf{x}}$ can be performed by the maximum differentiation competition [Wang and Simoncelli] or the perceptual attack [Zhang et al.].
>
> The reviewer is right that the term "sensitivity" was not precise. We modified it to "directional sensitivity", since it determines the sensitivity of the system for a particular point (original image) in a particular direction (distorted image).
>
> We rephrased the manuscript accordingly. Given those clarifications, the reviewer's question on maximum differentiation can be seen in a different light. For a given image and direction of distortion, in the definition of sensitivity, $S(\mathbf{x}, \mathbf{\hat{x}}) = \frac{D_p(\mathbf{x}, \mathbf{\hat{x}})}{||\mathbf{x} - \mathbf{\hat{x}}||_2}$, one already fixed a direction. Therefore, by definition, one cannot "look for" alternative directions since they would correspond to sensitivities defined for other points.
>
> In short, Maximum Differentiation works with locus of constant distance, not locus of constant sensitivity. Instead, what can be done is, given the directional sensitivity (or derivative) of one metric, find the direction that for a given Euclidean distance $||\mathbf{x} - \mathbf{\hat{x}}||_2$ in which S is maximum (or minimum), and then compare it with the behaviour of another perceptual metric (or a human observer). This is interesting to evaluate the proposed models in extreme directions.
>
> Certainly the analysis of the behaviour in particular directions is really interesting and actually it is an ongoing work we have right now. But this manuscript is focused on explore the general (direction independent) behaviour, by taking random directions in spheres of fixed Euclidean radius. Note that particular directions like luminance increase/decrease, contrast increase/decrease, or even distortions like compression, are very special (compared to the infinite directions that one could explore). This opens many options like which directions to explore, what the most interesting directions are, how to combine different directions, designing specific directions with MAD-like or adversarial attack methods (as suggested by the reviewer), analyse the shape of a surface of constant sensitivity, etc. Therefore we felt that this should be a separate (although related) work.
>
> > The authors should clearly state the meaning of $p(\mathbf{x})$: is it probability density function (PDF) or probability mass function (PMF)? Working with PDF is less reasonable, if the learned distribution is not smooth.
>
> We use PDF as this is the term given by the PixelCNN++ authors. While there is a discretisation step in the PixelCNN++ architecture for computational purposes the model admits continuous inputs and provides soft outputs.
>
> It is true that there is no way to ensure that the model does not have singularities, it is expected to have a soft response since the activation functions are ELU. In a new section in the appendix we explore the smoothness of the output, while there is a discretisation in the outputs it is only noticeable at extremely low levels.
>
> > Perturbing images with additive uniform noise makes the results in this paper less interesting. Any ideas on the optimal perturbation for the same goal (using probability-related factors to explain/predict human perception)?
>
> This is related to the first question. There are particular distortion directions that affect perception in a special way, variation of luminance or contrast in certain frequencies are examples. However, as we want to capture the general behaviour, we sampled uniformly on the sphere at a constant distance from the image. It is not a particular noise but a way of sampling on the sphere, other options like sampling in a grid around the sphere are unfeasible given the dimensionality of the data.
>
> The impact of variations of probability in a specific direction on (metric-based) sensitivity can be compared to (actual human) sensitivity. This interesting comparison can be done by measuring the human transduction functions mentioned by reviewer 1, as now included in the discussion as necessary further work to confirm the trends revealed in this work.

---

> ### Author Response · Authors · 2023-11-16
> **Rebuttal for Reviewer Cqjc Part 2**
>
> > Parameteric prediction in Eq. (3) and Eq. (4) can be trivial and thus meaningless with a deliberately chosen set of $\hat{\mathcal{X}}_1=\{ \hat{\mathbf{x} }\}$. Putting another way, it is not hard to come up with another $\hat{\mathcal{X}}_2=\{ \hat{\mathbf{x} }\}$ to make the parametric prediction nearly impossible.
>
> We do not fully understand the fourth question: in principle, for any given image/distortion the selected generative model can always be used to compute the probabilities, and then Eqs. 3 and 4 can always be applied. Would it be possible for the reviewer to provide additional clarification on the question, please?
>
> > How to apply the computational analysis in the paper? Can the results reflect which quality metrics are better explaining human perception?
>
> We are not addressing which metric is better explaining human perception, but how is their relation to probability. If we get a model for this relation in the case of perceptual metrics, we assume that this model will also be approximately valid for actual human perception.
>
> The outcome from the study is that all the considered metrics are similarly related to probability. Please look at the new section in the appendix where a unique simple model can predict reasonably well the sensitivity of all the analysed metrics.
>
> This implies that beyond the different nature of the metrics (some are related to physiology, while others are pure fits to subjective distortion data) all have the same relation to probability. This s a strong suggestion that actual human sensitivity will be related to probability in a similar way. Therefore, probability-related factors may explain up to 80\% of the variability of sensitivity.
>
> > The reviewer fails to understand the message of Fig. 2.
>
> We have added a description on what can be gained from Figure 2. Besides we have rewritten part of the section 3.1. We mist to clarify histograms were computed using the data in the previous section.

---

### Official Review · Reviewer_6JJg · 2023-10-30

**Soundness:** 2 fair
**Presentation:** 2 fair
**Contribution:** 3 good
**Rating:** 5
**Confidence:** 4

**Summary:**

The authors investigate the relationship between image-probability factors and four different proposed measures of human sensitivity based on image-quality measures. They use regression analysis and mutual information to quantify what is shared between the probability factors and the sensitivity measures. They find that $log(p(\tilde{x}))$ is most indicative of the perceptual sensitivity for the tested distances.

**Strengths:**

The paper is topical given current widespread interest in generative models capturing probability distributions. The introduction and background section are well written and cover a lot of interesting ground connecting classical work in neuroscience and perception to more modern computational models. The authors also test a variety of different proposed perceptual distance measures.

**Weaknesses:**

W1: The paper starts out (with the title) stating that it is investigating the link between image statistics and human perception, however, there is no human perception actually studied in the paper. The authors mention this in the discussion as a limitation (that as a proxy for human perception, perceptual metrics were used). However, all of these perceptual metrics are known to be imprecise. Given this, it is overall difficult to know what to take away from the paper. Are the authors actually studying human perception, or are they studying properties of “perceptual distances” that have previously been described? Additionally, the wording throughout the paper should be softened to make it clear that these are just distances measured by a model and not human measurements.

W2: The classical psychophysics experiments are difficult to follow. Perhaps a schematic would help readers understand what is actually being tested in the models? Additionally, the paper states that the classic psychophysics experiments are an “independent way” to validate the proposed models. I’m not so sure that this is actually independent, as presumably, many of the developed distance measures take into account (either explicitly or implicitly) visual sensitivity based on luminance, contrast, and special frequency.

W3: The authors use a model trained on the CIFAR dataset to get p(x) and then use this for testing, but I’m not sure that this accurately captures relevant properties of human perception (and also whether the distances measures that they are studying are valid on this dataset). Discussion about the distribution mismatch between the datasets that were used to test models of distance measures and the dataset used for training the model used to obtain p(x) might be beneficial.

**Questions:**

Q1: In the first paragraph of 2.1 the authors have a sentence saying, “This ratio is big at regions of the image space where human sensitivity is high and small at regions neglected by humans.” This seems a bit opaque to me. What are “regions of image space”? Additionally, it seems like there need to be constraints of (1) small perturbations since this is a local measurement and (2) comparing these “regions of image space” only when the ||x-\tilde(x)||^2 is equal between the two tested regions. Are these necessary?

Q2: In the first sentence of 3.1 the authors use the phrase “sensitivities of the metrics”. I think this just means something like “how these metrics change with different probability factors”? The wording is a bit confusing because the paper is studying “sensitivities."

Q3: Figure 2 is somewhat difficult to interpret. Clearly defining the x and y axes in the figure (rather than in the text) would help the reader.

Q4: Figure 4 y-axis is labeled as “Sensitivity” but it would be helpful to explicitly list this as something like “DISTS-derived Sensitivity”.

Q5: The authors state in the discussion that “images in the subjective experiments usually fall out of the range where you can use the current probability models”. Could the authors spell this out a little more?

---

> ### Author Response · Authors · 2023-11-16
> **Rebuttal for Reviewer 6JJg**
>
> Thank you for your review.
>
> > W1: The paper starts out (with the title) stating that it is investigating the link between image statistics and human perception, however, there is no human perception actually studied in the paper. The authors mention this in the discussion as a limitation (that as a proxy for human perception, perceptual metrics were used). However, all of these perceptual metrics are known to be imprecise. Given this, it is overall difficult to know what to take away from the paper. Are the authors actually studying human perception, or are they studying properties of “perceptual distances” that have previously been described? Additionally, the wording throughout the paper should be softened to make it clear that these are just distances measured by a model and not human measurements.
>
> The conceptual difference between (real) "human perception" and (proxy) "perceptual metrics" has also been raised by reviewer~1 (RwsQ). We think it is an important point: please have a look at the extensive response we gave above, that includes conceptual clarifications in the manuscript and new experiments.
>
> > W2: The classical psychophysics experiments are difficult to follow. Perhaps a schematic would help readers understand what is actually being tested in the models? Additionally, the paper states that the classic psychophysics experiments are an “independent way” to validate the proposed models. I’m not so sure that this is actually independent, as presumably, many of the developed distance measures take into account (either explicitly or implicitly) visual sensitivity based on luminance, contrast, and special frequency.
>
> We have included an schematic for the psychophysic experiments. We agree that the evaluation of the proposed models can be done using more elaborated psychophysic experiments, this is an ongoing task. However, we feel that the results shown in the manuscript really are an independent way of testing. None of the metrics used as a proxy are fitted neither designed to explicitly reproduce these psychophysic experiments. Therefore, if these metrics reproduce the experiments (and we agree they should do so until certain level) is because they are a good proxy for perception.
>
> > The authors use a model trained on the CIFAR dataset to get p(x) and then use this for testing, but I’m not sure that this accurately captures relevant properties of human perception (and also whether the distances measures that they are studying are valid on this dataset). Discussion about the distribution mismatch between the datasets that were used to test models of distance measures and the dataset used for training the model used to obtain p(x) might be beneficial.
>
> It is true that the images of the results are smaller than we wish. We are working on the limitation of the probabilistic model, but by now it seems that there is not a trustable one that operate with bigger images. Each measure was trained (and tested) on different datasets. For instance LPIPS was trained with 64x64 size images (not far from CIFAR size). The main effect can be the multiscale decomposition used either explicitly or implicitly in the subsampling process of the neural network architecture
>
>
> ## Questions
> Regarding the questions. We have introduced the recommended changes to the figures and clarified the language. Specifically, we have rewritten the entire explanation of the equation in section 2.1. We hope to make the definition and the intuition behind it clearer. We have rewritten section 3.1, in the previous version we did not specify that the histograms are calculated using the data presented in the previous section. This was important also in Figure 2, apart from defining the axis correctly. Note that Figure 4 is now computed with a generic set of coefficients (which are good to predict all the metrics).

---

> > ### Comment · Reviewer_6JJg · 2023-11-22
> > **Response to author rebuttal**
> >
> > Thank you for the clarifications. After reviewing the author's responses, updated manuscript, and other reviewers' comments I am inclined to keep my current score. In its current form, I think the manuscript is borderline for ICLR. In particular, I am still unsure that the authors have fully addressed the concern about human perception vs. perceptual metrics (for instance, what *is* the new title you are proposing? This seems important for reviewers to see before publication. But this is not my only concern.)

---

> > > ### Author Response · Authors · 2023-11-22
> > > **Response**
> > >
> > > The title we have in mind is 'Disentangling the Link Between Image Statistics and Perceptual Metrics’.
> > >
> > > Our motivation is having a valuable piece of work, and since the reviewer did the effort of reading the paper twice (which we really appreciate), it would be really helpful for us if you could specify the concerns. We are committed to publish the work here or elsewhere and we would like to solve all the concerns.

---

### Official Review · Reviewer_Y3LV · 2023-11-01

**Soundness:** 3 good
**Presentation:** 3 good
**Contribution:** 3 good
**Rating:** 6
**Confidence:** 3

**Summary:**

This paper directly evaluates image probabilities using a generative model PixelCNN++ and analyzes how probability-related factors can be combined to predict human perception via the sensitivity of SOTA image quality metrics. Further, it uses information theory and regression analysis to find a combination of just two probability-related factors that achieve a high correlation with the SOTA image quality metrics. Finally, this probability-based sensitivity is psychophysically validated by reproducing the fundamental trends of the Contrast Sensitivity Function, its suprathreshold variation, and trends of the Weber law and masking.

**Strengths:**

An interesting study on "Disentangling the Link Between Image Statistics and Human Perception" with experimental verification.

**Weaknesses:**

None

**Questions:**

This is an exciting study.  I did not notice major defects in this manuscript, to my knowledge. However, it would be more interesting if there could be more experimental verifications on other SOTA IQA metrics. And what will happen if more accurate probabilities are estimated by more advanced generative models?

The impact of this work would be increased by providing the source code.

$log (p(\hat{\mathbf{x}}))^\gamma, log (p(\hat{\mathbf{x}}))^2, log (p(\hat{\mathbf{x}}))^{-1}, ...$ should be $\left(\log p(\mathbf{x})\right)^\gamma, \left(\log p(\mathbf{x})\right)^2, \left(\log p(\mathbf{x})\right)^{-1}, ...$

To be self-contained, symbols in Table 1, e.g., $B, \mu, \Sigma$, can be explained in place.

The editing can be improved, e.g., log -> \log; Figure -> Fig.; table -> Table; section  -> Sec.; Eq.  \ref{} -> Eq. \eqref{}; overlapped terms in Fig. 3; the period in Appendix B, C, D;  [0’3,0’2,0’1,0,1,2,3] in Table 4; and the presentation quality of most of the Figures in the manuscript.

---

> ### Comment · Reviewer_Y3LV · 2023-11-14
> **Decrease my rating**
>
> After reading the comments from other reviewers, especially the Weaknesses part, I decided to decrease my rating.

---

> ### Author Response · Authors · 2023-11-16
> **Rebuttal for Reviewer Y3LV**
>
> Thank you for your review. We have made the suggested text changes and will release source code on acceptance. We have also made the recommended changes to formatting and the figures.
>
> Regarding more advanced generative models, we require these generative models be able to produce accurate estimates of the PDF, rather than just being able to sample from them. Generative models that satisfy this constraint are mainly normalising flows, however these are usually trained on a restricted subset of images (e.g. faces), that do not capture the properties of natural images as a whole. We look forward to new generative models that satisfy these properties as well as being able to use larger images.
>
> Please take a look at our responses to the weakness presented by the other reviewers, and consider raising the score again.

---

> > ### Comment · Reviewer_Y3LV · 2023-11-22
> >
> > Thank you for your response. I will keep the current positive rating.

---

### Official Review · Reviewer_RwsQ · 2023-11-03

**Soundness:** 2 fair
**Presentation:** 2 fair
**Contribution:** 2 fair
**Rating:** 5
**Confidence:** 4

**Summary:**

The authors propose to further test the relation between image statistics and perceptual sensitivity. To this purpose they propose to test and to compare several previously proposed heuristic models for predicting perceptual sensitivity (and also combination of those models). The main contribution is the use of deep neural network architectures to provide a direct estimate of the distribution of natural images. Finally, the authors validate their approach by reproducing classical psychophysical functions.

**Strengths:**

- the work is well-grounded in the theoretical vision science field with sufficient references to previous research
- extensive model comparison (several predictive models vs several perceptual distances)
- validation on empirical data

**Weaknesses:**

**Major**

- Methodological issues : (i) The paper does not really tackle the question of image statistics and human perception as in fact human perception is replaced by perceptual distances which are only computational models that mimic human perception.
(ii) The use of polynomial combinations of different models does not really make sense in the proposed work. Polynomials can often fit any data so they are not really falsifiable... Here it is true that the authors limit themselves to order 2 polynomials but the decision is only based on fit quality. Is there any reason/motivation to get a second order polynomial beyond fit quality ?

- Here the authors seem to avoid assuming that there is an underlying transduction function proper to an observer. I think this could be a strength but the authors do not mention this and recent relevant literature is not cited (see below). When you assume the existence of a transduction function (that is actually measurable in an observer) and with extra optimal coding assumption you can explicitly derive the relation between the probability density and the perceptual distance. Though this framework is somehow more restricted because it requires assumptions about the nature of the image distortion. In contrast, in the proposed work it should be valid for any distortion (as long as it is small enough) but only adding uniform noise is tested...

Extra-literature :
- Wei, X. X., & Stocker, A. A. (2017). Lawful relation between perceptual bias and discriminability. Proceedings of the National Academy of Sciences, 114(38), 10244-10249.
The MLDS technique to measure transduction functions and some use cases:
- Knoblauch, K., & Maloney, L. T. (2008). MLDS: Maximum likelihood difference scaling in R. Journal of Statistical Software, 25, 1-26.
- Charrier, C., Knoblauch, K., Maloney, L. T., Bovik, A. C., & Moorthy, A. K. (2012). Optimizing multiscale SSIM for compression via MLDS. IEEE Transactions on Image Processing, 21(12), 4682-4694.
- Vacher, J., Davila, A., Kohn, A., & Coen-Cagli, R. (2020). Texture interpolation for probing visual perception. Advances in neural information processing systems, 33, 22146-22157.

**Minor**
- Throughout the paper, it is unclear what is the prediction of $S$ from the probabilistic factor and it makes figure 2 hard to understand. Why are those histograms useful ? Could we expect to measure such an histogram in a human observer ? Indeed what would really be good to see in this figure is a row corresponding to human observer.
- It is unclear how the probability of a natural image is computed from PixelCNN++ ... It is not straightforward and the authors should not assume that the reader is familiar with any neural network...
- Where are the real data in Figure 4 ? This would be useful for a reader who does not know those curves...

Post discussion update : score raised to 5.

**Questions:**

See above.

---

> ### Author Response · Authors · 2023-11-16
> **Rebuttal for Reviewer RwsQ**
>
> We thank you for your review.
>
> > Methodological issues : (i) The paper does not really tackle the question of image statistics and human perception as in fact human perception is replaced by perceptual distances which are only computational models that mimic human perception.
>
> The first comment about the conceptual difference between "perception" vs "perceptual metrics" is totally fair. It is true that the current title does not reflect these differences properly (something easy to fix), but, by no means we tried to mislead the reader on this. In the camera ready version we will clarify the title slightly. We agree with the reviewer: note that many times (from the abstract, introduction, and the further work pointed out in the discussion) we say that metric sensitivities are *just approximations* of actual sensitivities, and the relations found here for all images/directions can/should be confirmed by psychophysical measurements (in specific images/directions).
>
> However, note that (as now better stressed in the introduction and discussion), relations with probability found in (debatable) metrics are still relevant in the endeavour of understanding human vision because (a) they can be used to check the relation at many points/directions at the same time (which is not feasible with actual experiments), and (b) they can be used to identify relevant points/directions to focus the actual experiments and get more conclusive results.
>
> Interestingly, a similar criticism also applies to the difference between "physiological/psychophysical metrics" vs "pure regression metrics": while the first are proper models of the actual visual system, the second are just convenient (but debatable) proxies.
>
> However (according to additional experiments included in a new appendix), when it comes to analysing the behaviour of the models in terms of *other factors* (e.g. the probability) the difference between "physiologically-inspired models" and "proxies" is not that important. Note that the relation between the probability and *all* the metrics is very similar because using a single equation (and single set of coefficients) does not degrade substantially the correlations obtained with different specific equations.
>
> The fact that a very simple model with a fixed set of coefficients is good to predict simultaneously the sensitivity of several perceptual metrics of very different nature (different plausibility) suggests that the analysis of "proxies" can be useful to understand the "real problem" as well.
>
> > (ii) The use of polynomial combinations of different models does not really make sense in the proposed work. Polynomials can often fit any data so they are not really falsifiable... Here it is true that the authors limit themselves to order 2 polynomials but the decision is only based on fit quality. Is there any reason/motivation to get a second order polynomial beyond fit quality ?
>
> We use polynomial combinations of different probability factors. Different exponents have been proposed in the literature. In our case we evaluate multiple exponents (the ones proposed in the literature and multiple others) and combinations of them. We selected the polynomial order 2 since including higher orders did not increase correlation, and if we limit to order 2 then the model is more interpretable. By looking at figure 2 it is clear that there is some non-linear relation between sensitivity and probability, so exponents different from one were the simplest choice. Also note that in this case polynomials can not fit the data if the features used (probability factors) are not related with sensitivity, even using polynomials of arbitrary large degree.

---

> ### Author Response · Authors · 2023-11-16
> **Rebuttal for Reviewer RwsQ Part 2**
>
> > Here the authors seem to avoid assuming that there is an underlying transduction function proper to an observer. I think this could be a strength but the authors do not mention this and recent relevant literature is not cited (see below). When you assume the existence of a transduction function (that is actually measurable in an observer) and with extra optimal coding assumption you can explicitly derive the relation between the probability density and the perceptual distance. Though this framework is somehow more restricted because it requires assumptions about the nature of the image distortion. In contrast, in the proposed work it should be valid for any distortion (as long as it is small enough) but only adding uniform noise is tested...
>
> The experimental transduction functions mentioned by the reviewer certainly have the advantage (over computational metrics) that they can be measured in humans and hence they directly represent perception (as opposed to proxies). The literature we cited in section 2.2.i and 2.2.ii connect this transduction with PDF equalisation. We thank the reviewer for leading us now to Wei \& Stoker 17, which we have included in Section 2.2 because it is based on the cumulative density as well.
>
> However, univariate approaches like the classical Laughlin 81 or McLeod \& Twer 01 are limited to specific directions of the image space, and this is a fundamental limitation, as acknowledged by the reviewer. Multivariate sequels (Malo\&Gutierrez 06...) are unbearably slow because require many samples to be trusted in high dimensions. Therefore, even though these approaches provide nonlinearities that can be directly compared to experimental transduction (always in specific directions), faster estimates of the PDF and of the human sensitivity are required to look for a probabilistic model valid at many points/directions simultaneously.
>
> In this unfortunate scenario where speed is the bottleneck, proxies of actual probability and actual sensitivity have to be considered. We are well aware that the experimental transduction function mentioned by the reviewer is approximated in this work by the nonlinear increase of distance (as measured by perceptual metrics) when one departs from a reference stimulus.
>
> This point (relation between nonlinear variation of distance and experimental transduction) which had not been explicitly stated in the text, has now been described in the definition of sensitivity (section 2.1) and in the discussion of Fig.2 (section 3.1).
>
> > Throughout the paper, it is unclear what is the prediction of $S$ from the probabilistic factor and it makes figure 2 hard to understand. Why are those histograms useful ? Could we expect to measure such an histogram in a human observer ? Indeed what would really be good to see in this figure is a row corresponding to human observer.
>
> As noted by the reviewer the dependence of sensitivity with different probability factors illustrated in Fig. 2 can be related to experimental transduction in humans because the sensitivity is the derivative of the variation of the distance in certain direction. This variation of distance is the computational model of the experimental transduction.
>
> As now noted in the discusion of Fig. 2, positive correlations of sensitivity with $\log(p(\mathbf{x}))$ or negative correlations of sensitivity with $\sigma(\mathbf{x})$ are consistent with saturating transductions when going from low-contrast images (high probability and low variance) to high-contrast images (low probability and high variance). Certainly, including a row corresponding to a human observer would be greatly beneficial. However, it would entail several intricate experiments that are beyond the scope of this publication.
>
> > It is unclear how the probability of a natural image is computed from PixelCNN++ ... It is not straightforward and the authors should not assume that the reader is familiar with any neural network...
>
> Thanks for pointing this out. We included some extra information on PixelCNN++ in section 2.5.
>
> > Where are the real data in Figure 4 ? This would be useful for a reader who does not know those curves...
>
> Regarding experimental figures, we *will* include (with proper scaling) the contrast-dependent CSFs of Georgeson \& Sullivan 75 and the sensitivities in terms of contrast can be related with the inverse of contrast incremental thresholds (e.g. Legge \& Foley 80 or Foley 94).

---

### Meta-Review · Area_Chair_6uvD · 2023-12-10

**Metareview:**

The described work attempts to further elucidates the link between the input statistics of the visual information that enters the human visual system and the sensitivities with which humans can discriminate said information along different dimensions. The innovation consists in using generative models for natural images as an implicit description/representation of natural image statistics, and perceptual image quality algorithms as proxies for human visual sensitivities. This, in principle, allows a more thorough test of the efficient coding hypothesis than what was possible with previous approaches operating on low dimensions (both stimulus and representation).

Reviewers generally appreciated the relevance of the research. However, they also raised concerns that the chosen proxies (on both ends) may represent approximations that are too coarse and diminish the strength of the conclusions when applied to human perception. The authors did show some predictions of their approach for some well-known human perceptual behavior (the contrast sensitivity function and weber's law; see Fig. 4) but a direct comparison to human data is not provided. Furthermore the presentation of these predictions makes it really difficult to judge their quality (e.g. weber's law predicts a constant relative sensitivity level).

Other concerns were related to more technical questions. Overall, confidence in the technical soundness of the paper was only moderate (avg score: 2.25). The common sense was that although this paper is proposing an interesting take on a relevant question, it is just a bit below threshold for ICLR.

**Justification For Why Not Higher Score:**

The proposed proxies may just be too coarse, and this uncertainty substantially diminish the strength of the approach. The presented results are not convincing enough.

**Justification For Why Not Lower Score:**

N/A

---

### Decision · Program_Chairs · 2024-01-16

Reject